# Reconstruction of Range-Doppler Map Corrupted by FMCW Radar Asynchronization

**DOI:** 10.3390/s23125605

**Published:** 2023-06-15

**Authors:** Kyung-Min Lee, In-Seong Lee, Hee-Sub Shin, Jae-Woo Ok, Jae-Hyuk Youn, Eung-Noh You, Jong-Ryul Yang, Kyung-Tae Kim

**Affiliations:** 1Department of Electrical Engineering, Pohang University of Science and Technology, 77 Cheongam-ro, Nam-gu, Pohang 37673, Republic of Korea; lovesw@postech.ac.kr; 2Department of Electronic Engineering, Yeungnam University, 280 Daehak-ro, Gyeongsan 38541, Republic of Korea; dldlstjd0322@yu.ac.kr; 3LIG Nex1 Co., Ltd., Yongin 13488, Republic of Korea; heesub.shin2@lignex1.com (H.-S.S.); jaewoo.ok2@lignex1.com (J.-W.O.); jaehyuk.youn@lignex1.com (J.-H.Y.); enyou76@lignex1.com (E.-N.Y.); 4Department of Electrical and Electronics Engineering, Konkuk University, 120 Neungdong-ro, Gwangjin-gu, Seoul 05029, Republic of Korea; jryang@konkuk.ac.kr

**Keywords:** frequency-modulated continuous wave, bistatic radar, synchronization

## Abstract

Frequency-modulated continuous wave (FMCW) radar system synchronization using external clock signals can cause repeated Range-Doppler (R-D) map corruption when clock signal asynchronization problems occur between the transmitter and receiver. In this paper, we propose a signal processing method for the reconstruction of the corrupted R-D map owing to the FMCW radar’s asynchronization. After calculating the image entropy for each R-D map, the corrupted ones are extracted and reconstructed using the normal R-D maps acquired before and after the individual maps. To verify the effectiveness of the proposed method, three target detection experiments were conducted: a human target detection in an indoor environment and a wide place and a moving bike-rider target detection in an outdoor environment. The corrupted R-D map sequence of observed targets in each case was reconstructed properly and showed the validity by comparing the map-by-map range and speed changes in the detected target with the ground-truth information of the target.

## 1. Introduction

Frequency-modulated continuous wave (FMCW) radar is a technology that measures a target’s range and speed using continuous frequency-modulated signals. It transmits continuous radar signals, having linearly modulated frequency, and obtains the range and speed information of the target by implementing a deramping process on the received signals [1]. Additionally, it generates continuous range-Doppler (R-D) images and combines them to make R-D video (i.e., an R-D map sequence) to detect and track the target. Because FMCW radar is less complex than the pulse-based radar system, it can be utilized in small radar platforms and near-field target detection environments, such as the detection of automobiles and humans [1,2,3,4,5].

Recent studies have used the FMCW radar system for bistatic radar configuration in various research fields, including detection, classification, and synthetic aperture radar (SAR) imaging [2,3,4,5,6,7]. Bistatic radar is a radar technique that observes a target by transmitting and receiving electromagnetic radar signals in a spatially separated positions of the transmitter and receiver [8,9]. Unlike the conventional monostatic radar, it provides the bistatic radar cross-section (RCS) information of an observed target, which can be used to perform more improved target detection and classification [6,7,8,10,11,12]. In addition, because the bistatic radar configuration has a spatial diversity of transmitter and receiver positionings, it can resolve the spatial constraints arising from the monostatic radar and use different incidence and reflection angles to obtain bistatic RCS.

However, because the bistatic radar uses different reference signals generated by different local oscillators (LO) for FMCW radar transmission in the transmitter and deramping process in the receiver, the difference between the two reference signals can cause errors in the deramping process, making it difficult to generate the entire R-D map sequence. This phenomenon is called asynchronization between the transmitter and receiver; furthermore, a synchronization process for a bistatic FMCW radar between the transmitter and receiver must be preceded to obtain a complete bistatic R-D map sequence.

To solve the bistatic radar asynchronization problem, several studies have been conducted to induce the timing of the same reference signal generation by applying external clock signals to each transmitter and receiver [13,14,15,16]. These studies solve the asynchronization problem by applying the equal external clock signal to the LO of each transmitter and receiver to simultaneously generate the same reference signals. These methods of synchronizing the FMCW radar using external clock signals are useful to build bistatic FMCW radar systems because they enable the control of the entire radar system through the control unit and facilitate expansion into the multistatic radar system configuration [17]. Figure 1 shows the FMCW radar synchronization structure using external clock signals performed in this study.

However, when the FMCW radar synchronization using external clock signals is inconsistent with the internal clock signals of each transmitter and receiver, their reference signals fail to occur at the accurate time, resulting in an asynchronization problem. Figure 2 shows examples of the R-D maps, observing a human target, where the left side shows the 102nd R-D map and the right side shows the 103rd R-D map.

In the case of the 102nd R-D map, the target appears because the synchronization between the transmitter and receiver is performed correctly; however, in the case of 103rd map, target information cannot appear on the R-D image, because the synchronization has not been performed. This is due to the mismatch between the external clock signal and the internal clock signal in the transmitter and receiver, which weakens the power of the reference signals in the transmitter and the receiver, resulting in an abnormal deramping process. This phenomenon is repeated randomly whenever the external and internal clock signals are matched or mismatched, causing the R-D map sequences to become corrupted repeatedly. Therefore, to minimize the above R-D map corruption, the time difference between two LOs must be decreased. Various hardware improvement methods have been developed to this end [13,15,18,19,20]; however, they are restricted to being hardware solutions and cannot be applied to various radar measurement environments, such as counting people or automobile detection, owing to the increasing cost and complexity of radar systems.

In this paper, we propose a signal processing algorithm that reconstructs the loss of R-D map sequences owing to the FMCW radar’s asynchronization. First, to find the corrupted maps among all the R-D maps, their two-dimensional image entropies are calculated to extract the corrupted maps. Then, using the range and speed information of the normally acquired R-D maps adjacent to each corrupted map, the target range and speed on the corrupted map can be estimated using linear estimation. The novelty of the proposed method is that it presents a new paradigm by solving it from a signal processing perspective, not from an existing hardware perspective, and that it can be applied to various radar measurement environments. To demonstrate the effectiveness of the proposed reconstruction method, three cases of the target detection experiments were conducted, and these results show that the range and speed of the target formed in the reconstructed R-D map sequence were almost similar to those of the reference radar case implemented for the ground-truth information.

The remainder of this paper is organized as follows. Section 2 depicts the R-D map sequence structure and the corruption of the R-D maps of the FMCW radar. Section 3 describes how to extract the corrupted R-D maps and reconstruct them using the linear estimation method. Section 4 shows three experimental results of R-D map sequence reconstruction and the validity of the proposed method. Finally, Section 5 and Section 6 summarize the discussion and conclusions, respectively.

## 2. Corruption of R-D Maps of FMCW Radar Using an External Clock Signal

General FMCW SAR systems generate one-dimensional range profiles using the linear frequency-modulated (LFM) waveform. When using the sawtooth LFM signal, the transmit signal can be expressed as follows [1,21]: (1)$$\begin{array}{c} s\left(t\right)=rect\left(t/T\right)\exp\left\{j2\pi f_{c}t\right\}\exp\left\{j\pi K_{r}t^{2}\right\} \end{array}$$
where rect(·) denotes the rectangular window, *T* is time width of the FMCW signals, $f_{c}$ is the carrier frequency and $K_{r}$ is the chirp rate. When the transmit signal is received in the receiver after being hit by the observed target, it can be expressed as [1,21]
(2)$$\begin{array}{c} s_{r}\left(t\right)=A\cdot rect\left(\left(t-\frac{R}{c}\right)/T\right)\exp\left\{j2\pi f_{c}\left(t-\frac{R}{c}\right)\right\}\exp\left\{j\pi K_{r}{\left(t-\frac{R}{c}\right)}^{2}\right\} \end{array}$$
where *A* is the amplitude, involving the antenna gain, transmitting power and RCS of the target; *R* is the round-trip distance from the transmitter to the target to the receiver; and *c* is the speed of light. To obtain the value of *R*, the following deramping process should be performed [1,21,22]: (3)$$\begin{array}{c} s_{d}\left(t\right)=s\left(t\right)\cdot s_{r}^{*}\left(t\right)=A\cdot rect\left(\left(t-\frac{R}{c}\right)/T^{\prime }\right)\exp\left\{j\frac{2\pi f_{c}R}{c}\right\}\exp\left\{j\frac{2\pi K_{r}R}{c}t\right\}\exp\left\{j\pi K_{r}{\left(\frac{R}{c}\right)}^{2}\right\} \end{array}$$
where $T^{\prime }$ is the reduced time width because the rectangular windows of the two signals are inconsistent, owing to the time delay caused by *R*, resulting in the time width decreasing after low pass filter processing [1] as shown in Figure 3. In general, these variations in time width are negligible since they are much smaller than *T* ($T^{\prime }\approx T$). In Equation (Equation 3), the third phase term is called the residual video phase and is generally ignored or compensated [22]. The first phase term is called the Doppler phase, which can be used to estimate the target’s speed, and the second phase term is the beat frequency, which presents the *R*. After performing Fourier transform in Equation (Equation 3), a range profile on the frequency domain can be obtained.

In conventional monostatic FMCW radar systems, s(t) and $s_{r}\left(t\right)$ are generated from the same LO, having only a time difference of round-trip delay (R/c). However, when two signals are generated from different LOs, a time difference between them can occur owing to the asynchronization of the hardware, which can be expressed as follows: (4)$$\begin{array}{cc} s_{d,async}\left(t\right)=s\left(t-\Delta t_{async}\right)\cdot s_{r}^{*}\left(t\right) & \\ \;\;\;\;\;=A\cdot rect(\left(t-\frac{R}{c}\right)/T_{async}^{\prime })\exp\left\{j\frac{2\pi f_{c}R}{c}\right\}\exp\left\{j\frac{2\pi K_{r}R}{c}t\right\}\exp\left\{j\pi K_{r}{\left(\frac{R}{c}\right)}^{2}\right\} \end{array}$$
where $T_{async}^{\prime }$ is the reduced time width caused by the time asynchronization error, $\Delta t_{async}$. This affects the range profile formed by the Fourier transform process and interrupts the collection of information on the target range. If $\Delta t_{async}$ is larger than $T_{async}^{\prime }$, an overlapping interval between the two signals does not exist, and the deramping is not performed. Hence, when transmitting and receiving FMCW signals with two different LOs such as in bistatic FMCW radar, the asynchronization phenomenon must be considered to obtain complete range profiles.

As shown in Figure 4, the bistatic radar system uses different LOs, which generate the FMCW reference signals in the transmitter and the receiver. The transmitter transmits the reference signal to the target, and the receiver uses it to perform the deramping process, obtaining a one-dimensional range profile [1]. In this study, we adopt a synchronization method for bistatic radar systems which uses external clock signals to solve the asynchronization problem raised in Equation (Equation 4). When the LO of the transmitter receives the clock signal from the outside, it generates *N* consecutive FMCW signals in a single frame and transmit them. Then, the receiver obtains *N* range profiles used to estimate the speed of the target through the Fourier transform. The overall process can be represented as shown in Figure 5. Then, one R-D map can be obtained in a single frame. This process is repeated to generate continuous R-D images, that is, an R-D map sequence.

However, the FMCW signal formation’s deramping process cannot be fully performed if the reference signal is not generated at the exact timing, thus adversely affecting the *N* range profiles in one R-D map.

Figure 6 depicts the FMCW radar experiment in this study, using the external clock signal generated in the micro-control unit (MCU). In this experiment, the human target moves from 1 m to 3 m under the speed of 1 m/s in a repetitive manner. Figure 7 shows the example results of the corrupted R-D maps obtained in the above experiment, and see the Appendix A, for the full corrupted R-D map sequence.

As shown in Figure 7, the R-D maps are corrupted when asynchronization occurs. Because the deramping process for the received signals is not accurately performed at the receiver, the power of the deramped signals is weakened, resulting in the increase in noise in the R-D maps. This asynchronization phenomenon occurs repeatedly throughout the entire R-D map sequence, which deteriorates its quality. Therefore, to accurately use the R-D map sequence, a reconstruction process for the corrupted R-D maps is imperatively required.

## 3. Proposed Method for Corrupted R-D Map Reconstruction

This Section introduces the reconstruction algorithm for the corrupted R-D map damaged by the asynchronization, as discussed in Section 2. First, the corrupted R-D maps are extracted from the original R-D map sequence using image entropy and reconstructed by applying the linear estimation method.

### 3.1. Corrupted R-D Maps Extraction

To find corrupted parts among all the R-D maps, we used the image entropy parameter. Image entropy is a quantitative parameter that indicates the degree of disorder of image information, and it tends to increase when the image is unfocused, distorted or affected by noise. Generally, Shannon entropy is adopted to calculate the image entropy, as follows [23,24]: (5)$$T_{m}=-\sum_{v}\sum_{r}\left|I_{m}\left(r,v\right)\right|\ln I_{m}\left(r,v\right),$$
where *r* and *v* are the range and speed, respectively; *m* is index of the R-D map; and $T_{m}$ is the image entropy of the *m*-th R-D map.

In general, noisy R-D maps have high image entropy as calculated by (Equation 5) because the noise affects the entire image, resulting in a certain degree of disorder in the image [23,24]. Hence, the corrupted R-D maps, which are affected by noise, have higher image entropy than the normally generated ones, and the corrupted maps can be extracted using this characteristic. As shown in Figure 8, each image entropy of the R-D map is calculated using (Equation 5), and the indices of the corrupted maps can be separated by setting the appropriate threshold.

To accurately find normal R-D maps, it is important to set an appropriate threshold value. There are two methods of setting thresholds: one is selecting a constant value, and the other is selecting values considering the variation in the signal, called adaptive thresholding [25]. For target detection in more diverse environments, the adaptive thresholding method is useful, as it can adapt to variations in the image entropy of the R-D map sequence. Several adaptive thresholding methods have been studied, and they can be used for finding corrupted R-D maps regardless of the fluctuations in the image entropy [25,26,27]. In this article, we adopt an adaptive thresholding algorithm that uses moving averages and normalized histograms [25] to determine the m-th threshold values, $H_{m}$, as follows: (6)$$H_{m}={\bar{T}}_{m,L}+F_{ac}\cdot \sigma_{m,L}$$
where
(7)$$\begin{array}{c} {\bar{T}}_{m,L}=\frac{1}{L}\sum_{i=1}^{L}T_{m-i}\\ \sigma_{m,L}=\sqrt{\frac{\sum_{i=1}^{L}{\left(T_{i}-{\bar{T}}_{m,L}\right)}^{2}}{L}}, \end{array}$$
where *L* is the window length and $F_{ac}$ is the scale factor. Equation (Equation 7) shows the mean and standard deviation values of the windowed signal. By adjusting $F_{ac}$ in (Equation 6), the average level of the threshold value can be controlled, enabling its application to various environments having different image entropies. As a result, this algorithm can sufficiently find the corrupted R-D maps because it is robust to changes in image entropy, owing to the variation in the experimental environment.

### 3.2. R-D Map Reconstruction Using Linear Estimation

Each corrupted R-D map, extracted in the previous subsection, is reconstructed individually by finding and using the adjacent normal R-D maps before and after. As shown in Figure 9, assuming that an environment in which the range and speed of the observed target do not change rapidly, compared to the frame rate, the target information (range and speed) in the corrupted R-D map can be estimated using the target information within the adjacent normal map before and after. In this study, we estimate the target information in the corrupted R-D map using the linear estimation, as follows: (8)$$\begin{array}{c} r_{m+q}=r_{m}+\left(r_{m+p}-r_{m}\right)\frac{q}{p}\\ v_{m+q}=v_{m}+\left(v_{m+p}-v_{m}\right)\frac{q}{p} \end{array}$$
where *m* is an index of the previous normal R-D map, *p* is an index interval between the *m*-th and the subsequent normal R-D map and *q* is an index interval between the *m*-th normal R-D map and the corrupted R-D map (q<p). Using the linear estimation method in (Equation 8), all the corrupted R-D maps are recovered, and the reconstructed R-D map sequence can be obtained.

Figure 10 shows the overall processes of the proposed method.

## 4. Experimental Results

In this section, the results of the actual experiment to which the proposed R-D map reconstruction method is applied are presented, centering on the recovered R-D maps. To demonstrate the effectiveness of the proposed method, two cases of human target detection experiments and one bike-rider target detection experiment were conducted, and the range and speed information of the recovered target and the ground-truth information measured in the same environment were compared. We used the Texas Instrument AWR 1843 radar module, and the frequency spectrum of the radar used in this study was W-band. The detailed radar parameters are shown in Table 1. The sum of the time duration of all FMCW signals was 6.6 ms, sufficiently smaller than the frame repetition interval, which satisfies the duty cycle condition. The dynamic range of all the images is set to 40 dB.

Since each experimental environment receives not only target information but also indoor or outdoor noise information, an additional noise removal process was performed using the feature of concentration at zero Doppler points. In addition, because the distance between the transmitter and the receiver (i.e., baseline) is close, a directly received signal from the transmitter occurs, which appears as high values in the zero Doppler and zero range of the R-D map. Therefore, in order to obtain complete R-D map, the direct received signal was also filtered in the two experimental results.

### 4.1. FMCW Radar Experiment on a Near-Field, Indoor Human Target

First, the R-D map sequence of the indoor human target was reconstructed, and repeated round-trip motions were performed, as shown in Figure 6. The experimental environment was the same as that described in Section 2, Figure 6 and Table 1. External clock signals were generated from the MCU, and 150 R-D maps were used to make the R-D map sequence. The image entropy for each map was calculated to extract the corrupted ones among the R-D map sequence, as shown in Figure 11. Because all the corrupted R-D maps have relatively high image entropy and the normally generated maps have low values, the corrupted R-D maps can be extracted by setting a threshold value using moving average thresholding. In this experiment, *L* and $F_{ac}$ were set to 10 and 0.5, respectively.

In this experiment, the R-D maps in Figure 7 are the examples of the corrupted R-D maps, and their reconstructed R-D maps are depicted in Figure 12, and the Appendix A.

As a result, the human target signals, which were not shown in Figure 7, owing to the noise effect, clearly appear in the reconstructed R-D maps. This study reconstructed the R-D maps, as shown in Figure 7, along with the other corrupted R-D maps. As a result, the overall quality of the R-D map sequence was improved significantly.

To evaluate the proposed method in terms of the target detection, we acquired the R-D map sequence measured in the same human target and environment regarding the ground-truth and compared the target detection results (range and speed).

Figure 13 shows the map-by-map variation in the detected target’s range and speed for each R-D map sequence. The measured range and speed in the corrupted R-D maps are significantly out of the ground-truth values, but those measured in the reconstructed R-D maps are almost the same. Therefore, it is concluded that the proposed method accurately estimates the actual range and speed of the target.

### 4.2. FMCW Radar Experiment on Far-Field Human Target

To verify that the proposed method is valid regardless of location, an additional human target detection experiment was conducted in a wider place, as shown in Figure 14.

In this case, the human target was moving away from the radar from 1 m to 15 m, and the 300 R-D maps were obtained.

The calculated image entropy and adaptive threshold of each R-D map are depicted in Figure 15, where the *L* and $F_{ac}$ was set to 10 and 0.3, respectively. It also shows that the corrupted R-D maps have higher image entropy than the normal ones. In this experiment, the image entropy of the normal R-D maps gradually increases as the range between the human target and the radar increases; however, because the image entropy of the corrupted R-D maps is not reached, they can be extracted using the adaptive thresholding algorithm addressed in Section 3.

Figure 16 and Figure 17 are the examples of the corrupted and reconstructed R-D maps, respectively, measured in the second experiment (Appendix A). Because each R-D map is strongly affected by noise, the target signal is not visible in Figure 16; however, in the reconstructed R-D maps in Figure 17, the target signals are estimated, and the noise effects decrease. As in the previous experiment, the detected target’s range and speed in the corrupted R-D and reconstructed R-D maps and the ground-truth are compared for each R-D map, as shown in Figure 18.

The range and speed of the detected target in the corrupted R-D maps are rapidly deviating from the ground-truth values, but those in the reconstructed R-D maps are almost similar. Hence, as in the previous experiment, it also confirmed that the R-D map sequence, reconstructed by the proposed method, accurately estimates the actual target’s motion.

### 4.3. FMCW Radar Experiment on Bike-Rider Detection

Finally, we conducted an outdoor experiment which observed a bike-rider, which has a higher speed than the human target.

Figure 19 shows the bike-rider target and the outdoor experimental setting. Unlike the other cases, the observed target had a speed higher than the human target and had a different RCS. In addition, as the parked cars were located in an experimental environment, the FMCW radar could be subject to strong noise signal interference. The bike-rider approached in the direction of the radar, as depicted in Figure 19. In this case, 150 R-D maps were used, and *L* and $F_{ac}$ were set to 10 and 0.1, respectively.

Figure 20 illustrates the image entropy and adaptive threshold of each R-D map obtained via observation of the bike-rider target. It shows that the image entropy decreases as the target approaches the radar, and the image entropy increases repeatedly as a result of the corrupted R-D maps. Despite this, the calculated threshold value can separate the R-D maps with high image entropy values that can be considered as the corrupted R-D maps.

Figure 21, Figure 22 and Figure 23 are the examples of the corrupted and reconstructed R-D maps of the bike-rider target detection experiment (see Appendix A), and the detected target’s range and speed in the corrupted R-D and the reconstructed R-D map sequences, respectively. Like other experiments, the signal of the detected bike-rider target clearly appears in the reconstructed R-D maps, and the detected target’s range and speed are almost similar to the ground-truth values. These results demonstrate that the proposed method sufficiently reconstructs the corrupted R-D maps, even when detecting a fast target and in environments with a high level of noise.

## 5. Discussion

According to the results of the three real experiments shown in Figure 6, Figure 14 and Figure 19, it was confirmed that the R-D map sequences corrupted by the FMCW radar’s asynchronization can be reconstructed using the proposed method. Additionally, the range and speed of the detected target in the reconstructed R-D maps are almost similar to the ground-truth value, which demonstrates that the proposed method could be used for the target detection and classification studies using the R-D map sequence. Hence, this study shows the possibility and feasibility of the bistatic radar synchronization using external clock signals based on signal processing.

However, the proposed method is requires improvement. The R-D map reconstruction method uses linear estimation, under the assumption that the changes in the range and speed of the observed target are insignificant. However, in certain circumstances, the accuracy of the proposed method may be weakened when the changes in the range and speed of the target are greater than the frame repetition interval (inverse of frame repetition frequency) of the FMCW radar system. Therefore, to use the proposed method in more diverse experimental environments, a more general R-D map reconstruction method will be implemented. Some studies have proposed an image reconstruction method based on compressive sensing [28,29,30,31], which recovers blurred images based on the sparsity-driven signal processing. These studies show that the proposed method could be improved by applying such a method, which will be conducted in future works.

This study proposes a new paradigm that overcomes asynchronization phenomena occurring in bistatic FMCW radar systems using signal processing. Conventional bistatic radar synchronization methods concentrate only on improving hardware of radar or adding other equipment, which increases the radar system’s complexity and cost. In addition, if the hardware is operating incompletely, the hardware-based synchronization methods cannot be performed normally, leaving the asynchronization errors. Therefore, utilizing the proposed method will complement the limitations of hardware-based synchronization methods, and more accurate bistatic radar signals and R-D maps can be obtained.

## 6. Conclusions

In this paper, we propose a signal processing method to reconstruct the R-D maps corrupted through FMCW radar asynchronization when the external clock signals are used. First, the image entropy of each R-D map is calculated. Furthermore, maps having higher image entropy than the threshold, which are regarded as corrupted R-D maps, are found. Then, each corrupted R-D map is reconstructed by estimating the range and speed of the target within it via the linear estimation method, and then the reconstructed R-D map is put back into the original one to make a fully reconstructed R-D map sequence. The range and speed of the detected target in the reconstructed R-D maps are confirmed to almost match the ground-truth values. Therefore, the proposed method can accurately perform the R-D map sequence’s reconstruction.

However, as discussed in the previous Section 5, the proposed method requires additional study on the observation and R-D map reconstruction of targets having rapid changes in range and speed compared to the frame repetition interval, and the method could be improved by developing reconstruction methods to prepare for a more diverse target detection environment, which will be carried out in future works.

Additionally, a basic synchronization signal processing approach could be used, which is used in all the research fields using bistatic FMCW radar systems. Specifically, it can be extended and used in the areas that require precise radar signal acquisition, such as bistatic FMCW SAR.

## Figures and Tables

**Figure 1 sensors-23-05605-f001:**
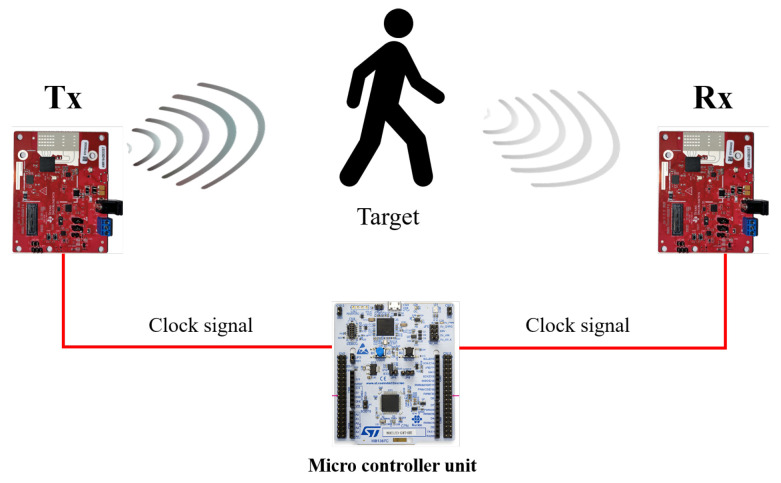
Bistatic FMCW radar synchronization using external clock signals.

**Figure 2 sensors-23-05605-f002:**
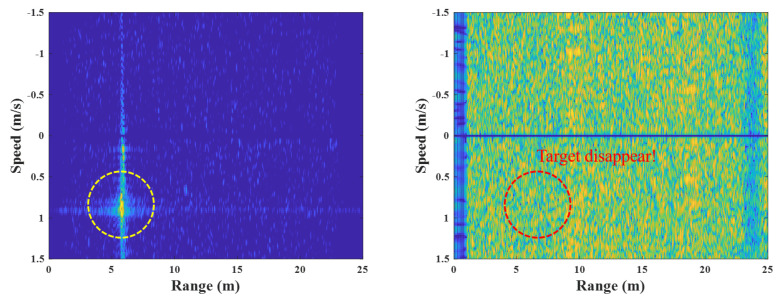
R-D map corruption owing to FMCW radar asynchronization.

**Figure 3 sensors-23-05605-f003:**
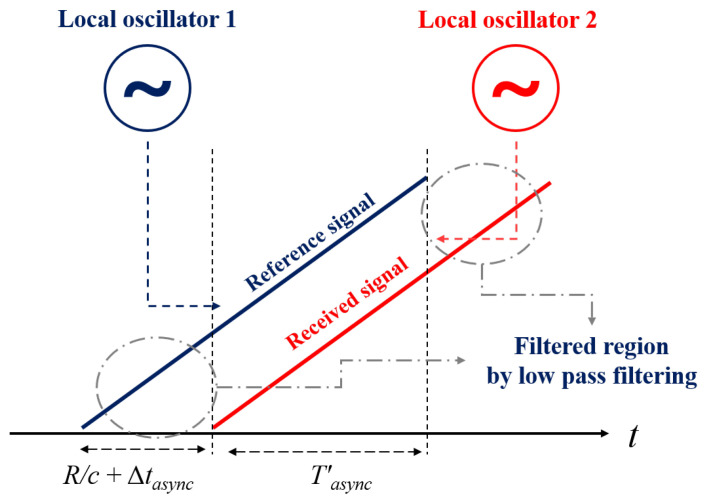
Deramping process of FMCW radar with asynchronization error.

**Figure 4 sensors-23-05605-f004:**
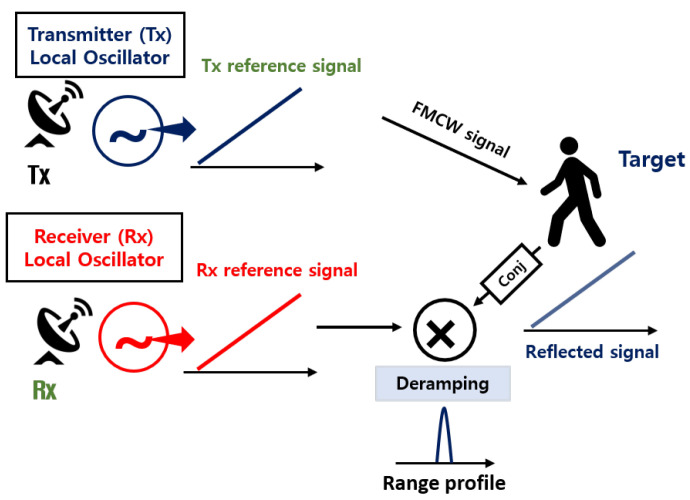
Geometry of bistatic FMCW radar.

**Figure 5 sensors-23-05605-f005:**
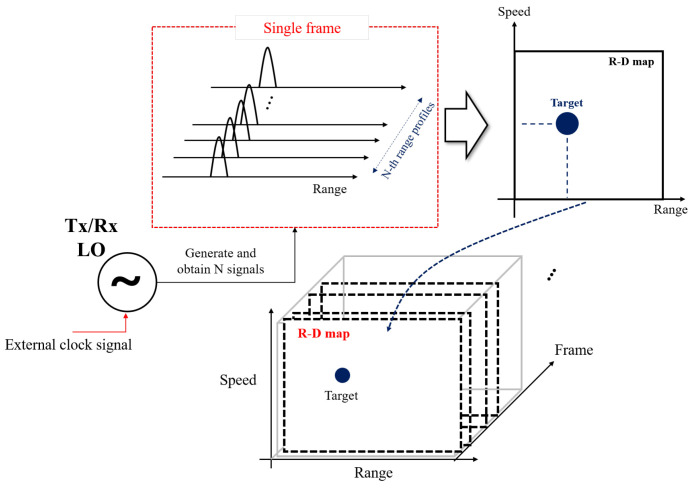
R-D map sequence composition.

**Figure 6 sensors-23-05605-f006:**
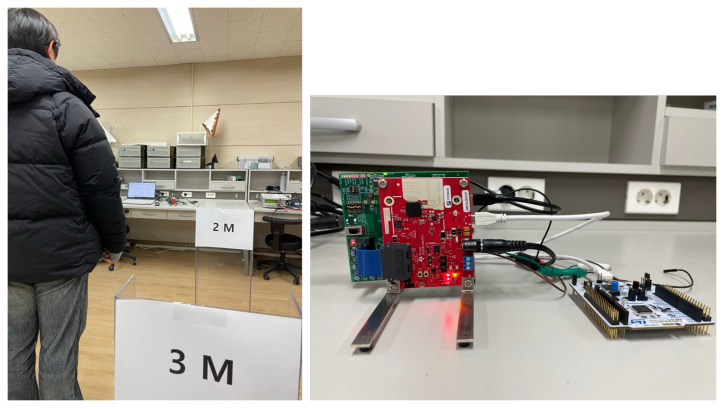
FMCW radar experimental environment and configuration (case 1).

**Figure 7 sensors-23-05605-f007:**
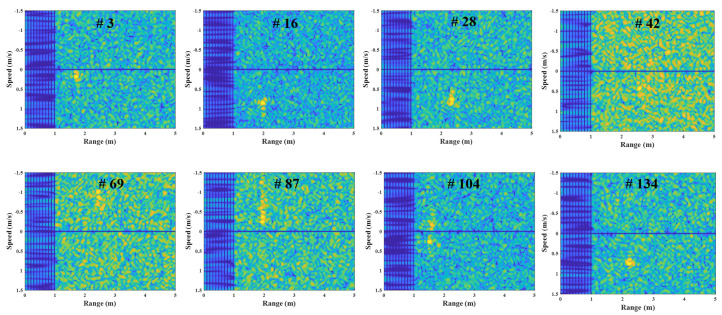
Corrupted R-D maps owing to the FMCW radar’s asynchronization (case 1, Appendix A).

**Figure 8 sensors-23-05605-f008:**
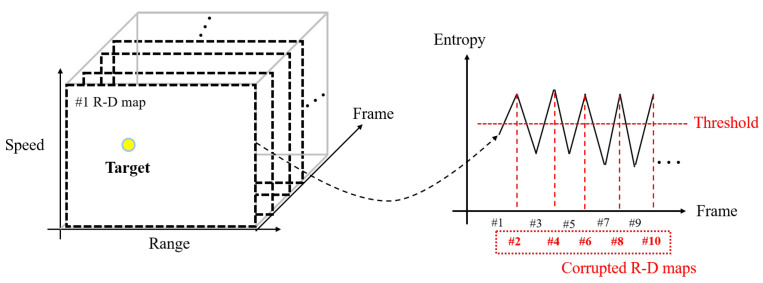
Corrupted R-D maps’ extraction using image entropy.

**Figure 9 sensors-23-05605-f009:**
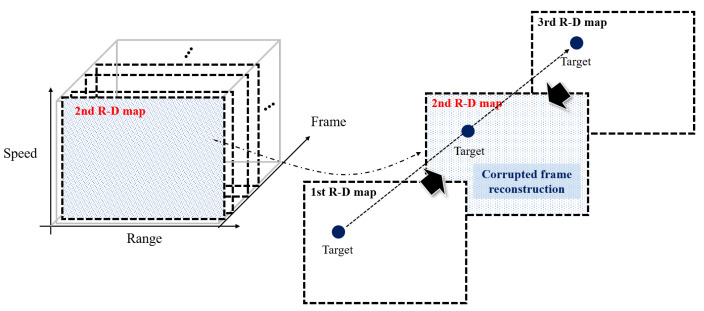
R-D Map reconstruction using linear estimation.

**Figure 10 sensors-23-05605-f010:**
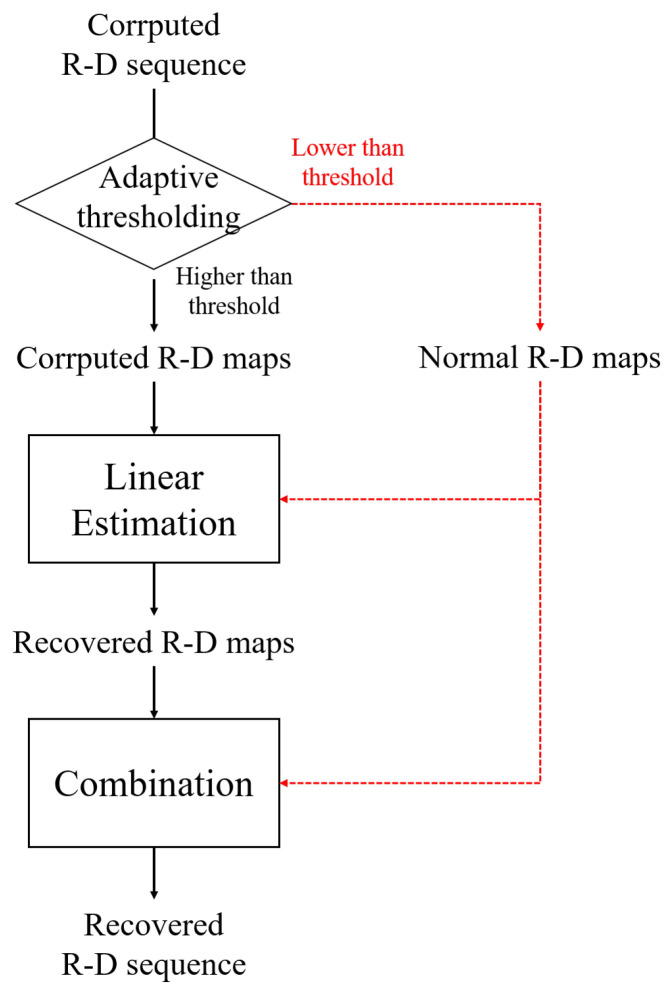
Flow chart of the proposed method.

**Figure 11 sensors-23-05605-f011:**
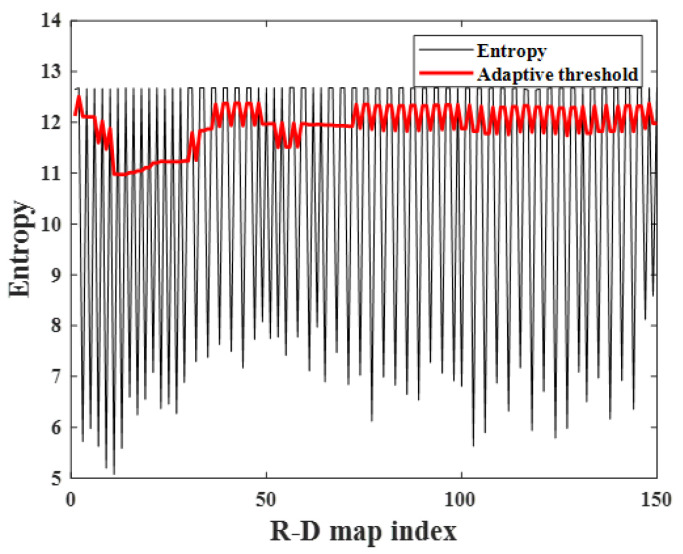
Image entropy variation with R-D maps (case 1).

**Figure 12 sensors-23-05605-f012:**
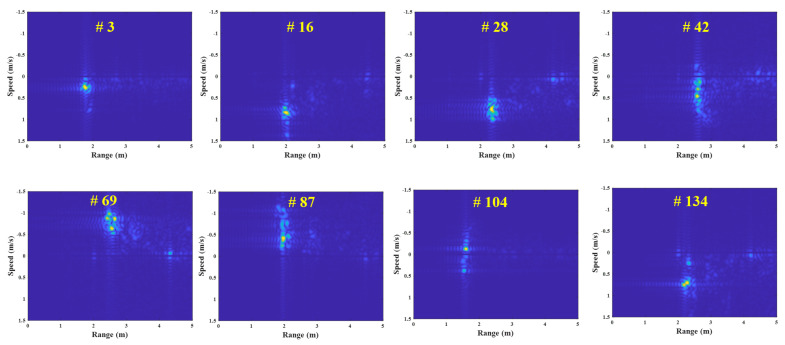
Reconstructed R-D maps using the proposed method (case 1, Appendix A).

**Figure 13 sensors-23-05605-f013:**
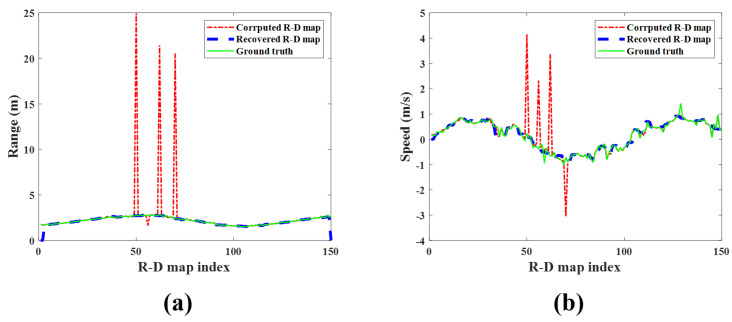
Detection results of the observed human target for each R-D map (case 1): (**a**) range and (**b**) speed. Green solid line: ground−truth; red dash−single dotted line: corrupted R-D maps; blue dashed line: reconstructed R-D maps.

**Figure 14 sensors-23-05605-f014:**
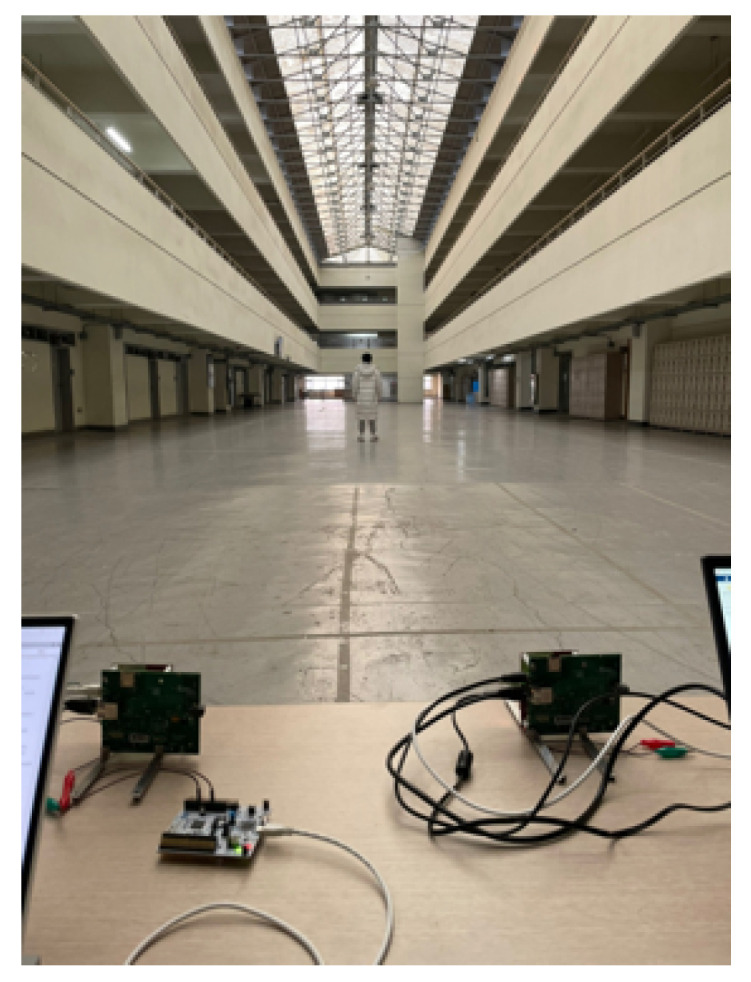
FMCW radar experimental environment and configuration in a wide place (case 2).

**Figure 15 sensors-23-05605-f015:**
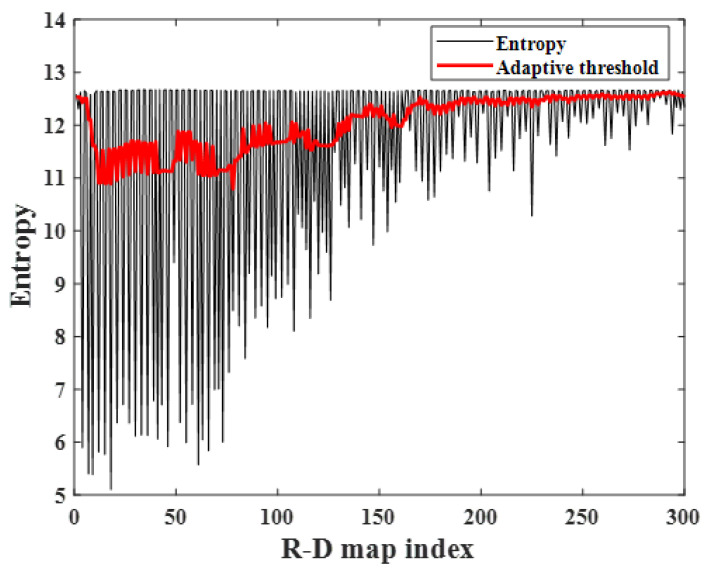
Image entropy variation with R-D maps (case 2).

**Figure 16 sensors-23-05605-f016:**
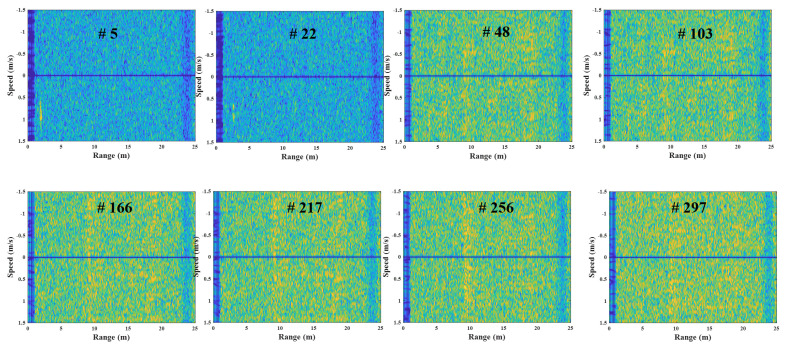
Corrupted R-D maps owing to the FMCW radar’s asynchronization (case 2, Appendix A).

**Figure 17 sensors-23-05605-f017:**
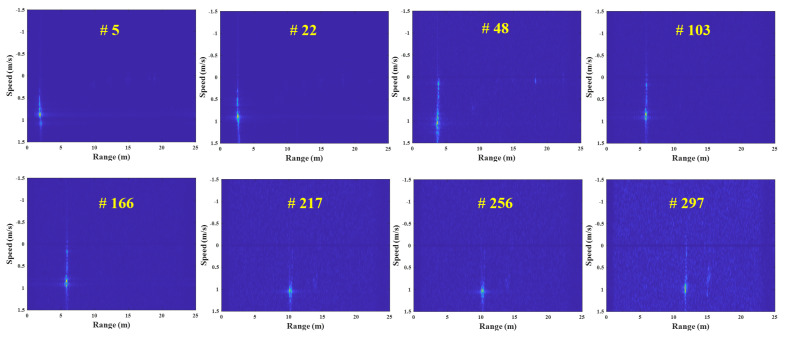
Reconstructed R-D maps using the proposed method (case 2, Appendix A).

**Figure 18 sensors-23-05605-f018:**
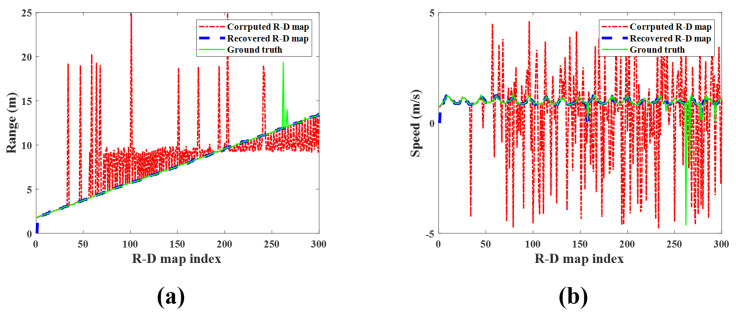
Detection results of the observed human target for each R-D map (case 2): (**a**) range and (**b**) speed. Green solid line: ground−truth; red dash−single dotted line: corrupted R-D maps; blue dashed line: reconstructed R-D maps.

**Figure 19 sensors-23-05605-f019:**
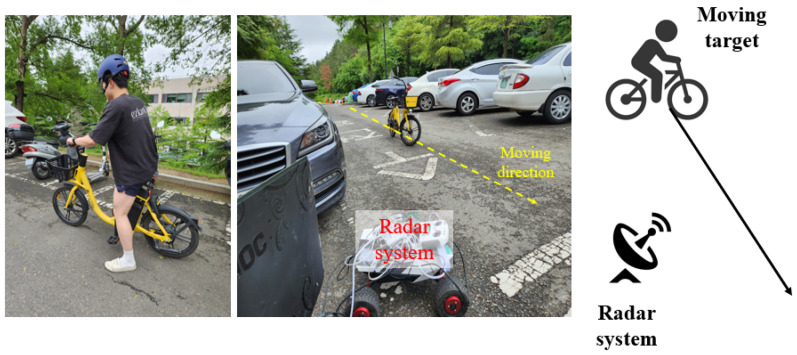
Outdoor FMCW radar experimental environment and configuration (case 3).

**Figure 20 sensors-23-05605-f020:**
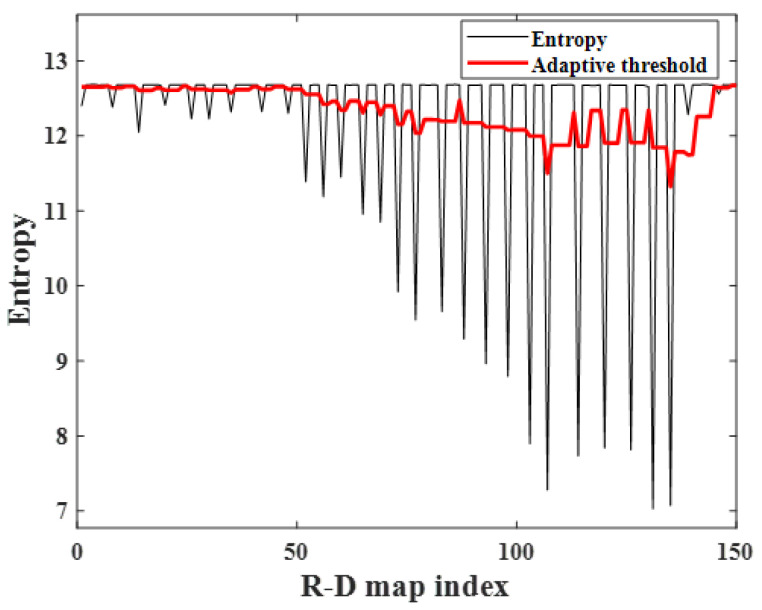
Image entropy variation with R-D maps (case 3).

**Figure 21 sensors-23-05605-f021:**
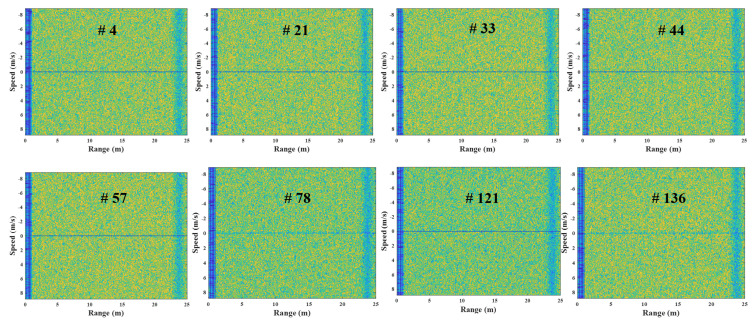
Corrupted R-D maps owing to the FMCW radar’s asynchronization (case 3, Appendix A).

**Figure 22 sensors-23-05605-f022:**
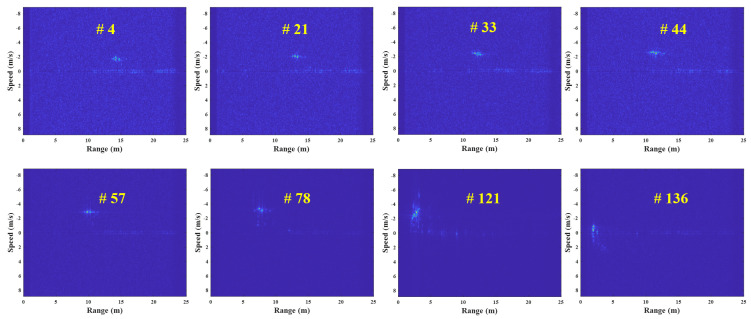
Reconstructed R-D maps using the proposed method (case 3, Appendix A).

**Figure 23 sensors-23-05605-f023:**
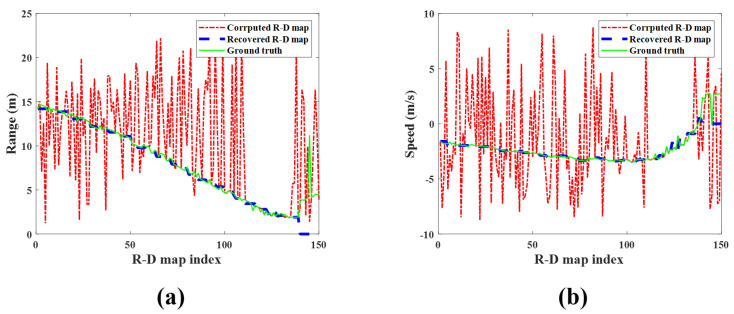
Detection results of the observed human target for each R-D map (case 3): (**a**) range and (**b**) speed. Green solid line: ground−truth; red dash−single dotted line: corrupted R-D maps; blue dashed line: reconstructed R-D maps.

**Table 1 sensors-23-05605-t001:** FMCW radar parameters.

Parameters	Value
Carrier frequency	77 GHz
Bandwidth	1.5 GHz
Chirp rate	29.98 MHz/µs
Frame repetition frequency	25 Hz
Time duration of a single FMCW signal	51.2 µs
Number of chirps in a single frame	128

## Data Availability

The data presented in this study are available on request from the corresponding author.

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
