# Peer review of "Reconstruction of Range-Doppler Map Corrupted by FMCW Radar Asynchronization"

_sensors, 2023, doi:10.3390/s23125605_

Round 1

Reviewer 1 Report

The topics discussed in the work are scientifically interesting and necessary. The article contains the state of the literature on the subject of the work. The article lacks literature references to some mathematical formulas, which makes it difficult for the reviewer to assess which mathematical formulas are original.

The editing page of the thesis is correct and legible. I did not find any significant editing or factual errors.

Reviewer 2 Report

This paper is well written. However I have following observation

1. Figure 17(b), The Ground truth color should be different to make better understand like 17(a)

1. Figure 12(b), The Ground truth color should be different to make better understand like 12(a)

Reviewer 4 Report

1. Firstly, for the bistatic radar, the measured bistatic range and angle could not translate into the exact location of the target unlike monostatic radar. This implies bistatic radar cannot be used for tracking, however, if only detection is required one can use the bistatic radar. Moreover, the detections from radar are noisy and inconsistent and difficult for analysis. For this, the trackers are used but as already mentioned bistatic radars are difficult to use for the tracking application. Therefore, please explain how the proposed scheme which is based on bistatic radar is useful in practical radar applications. 

2. In introduction the authors have mentioned: "In addition, because bistatic radar configuration has a spatial diversity of transmitter and receiver positioning, it can resolve the spatial constraints arising from the monostatic radar". Here, what type of spatial constraints the authors are referring  to? 

3. In introduction with 102th and 103th does authors means to say 102th frame and 103th frame? If that is the case the target is appear to be static as it is in same range cell. However, the speed scale is indicating otherwise and showing the speed to be 1m/s, please explain.

4. The statement made by the authors: "Then, using the range and speed information of the normally acquired R-D maps adjacent to each corrupted map, the target range and speed on the corrupted map can be estimated using linear estimation". This, implies that the target's radial range and radial velocity is changing linearly or constant with the frame. And the same is shown in Fig.  17. The linear transformation would not yield the accurate results for highly maneuvering targets, for that case I would recommend to perform simulations with highly maneuvering targets and use some nonlinear estimation.  

5. Also, how to evaluate the correctness of the linear estimation model in Equation 6. 

6. Please explain adaptive thresholding technique. 

7. The radar configuration values given in Table. 1 is not consistent. For instance the chirp rate of 29.98 MHz/micro seconds and 1.5Ghz bandwidth yields the pulse length of 0.5 microseconds. Subsequently, the frame rate of 25 Hz or 0.04 seconds of frame length yields 800 pulses in single frame. However the Table indicate otherwise (128 pulses in single frame). Please explain. 

Round 2

Reviewer 3 Report

My comments on the initial version of the manuscript have been sufficiently addressed by the authors in this revised version. I have no further comments on the technical aspects. The manuscript may be considered for publication after a proofreading.

Reviewer 4 Report

I have no further comments.